# Extracellular DJ-1 induces sterile inflammation in the ischemic brain

**Koutarou Nakamura**[1,2,3], **Seiichiro Sakai**[1,2], **Jun Tsuyama**[1,2], **Akari Nakamura**[1,2],
**Kento Otani**[1,2], **Kumiko Kurabayashi**[1,2], **Yoshiko Yogiashi**[1,2], **Hisao Masai**[3,4],
**Takashi Shichita**[1,2,5] *

1 Stroke Renaissance Project, Tokyo Metropolitan Institute of Medical Science, Tokyo, Japan, 2 Core
Research for Evolutional Science and Technology, Japan Agency for Medical Research and Development,
Tokyo, Japan, 3 Department of Computational Biology and Medical Sciences, Graduate School of Frontier
Sciences, The University of Tokyo, Chiba, Japan, 4 Department of Genome Medicine, Tokyo Metropolitan
Institute of Medical Science, Tokyo, Japan, 5 Precursory Research for Innovative Medical Care, Japan
Agency for Medical Research and Development, Tokyo, Japan

* shichita-tk@igakuken.or.jp

UNITED STATES

**Data Availability Statement:** All relevant data are
within the paper and its Supporting Information
files.

**Funding:** Leading Graduates Schools Program,
"Global Leader Program for Social Design and

## Abstract

Inflammation is implicated in the onset and progression of various diseases, including cere-
bral pathologies. Here, we report that DJ-1, which plays a role within cells as an antioxidant
protein, functions as a damage-associated molecular pattern (DAMP) and triggers inflam-
mation if released from dead cells into the extracellular space. We first found that recombi-
nant DJ-1 protein induces the production of various inflammatory cytokines in bone
marrow–derived macrophages (BMMs) and dendritic cells (BMDCs). We further identified a
unique peptide sequence in the αG and αH helices of DJ-1 that activates Toll-like receptor 2
(TLR2) and TLR4. In the ischemic brain, DJ-1 is released into the extracellular space from
necrotic neurons within 24 h after stroke onset and makes direct contact with TLR2 and
TLR4 in infiltrating myeloid cells. Although DJ-1 deficiency in a murine model of middle cere-
bral artery occlusion did not attenuate neuronal injury, the inflammatory cytokine expression
in infiltrating immune cells was significantly decreased. Next, we found that the administra-
tion of an antibody to neutralize extracellular DJ-1 suppressed cerebral post-ischemic
inflammation and attenuated ischemic neuronal damage. Our results demonstrate a previ-
ously unknown function of DJ-1 as a DAMP and suggest that extracellular DJ-1 could be a
therapeutic target to prevent inflammation in tissue injuries and neurodegenerative
diseases.

## Introduction

Inflammation is implicated in the pathophysiology of a wide range of illnesses, including neu-
rological diseases and psychiatric disorders [1]. Circulating or tissue-resident immune cells
trigger inflammation upon encountering stimuli such as infections and tissue damage [2].
During infections, pathogen-specific compounds induce the production of various inflamma-
tory mediators in innate immune cells through the activation of pattern recognition receptors
(PRRs) [3]. PRRs are also activated by endogenous "self" molecules released into the

Management" by the Ministry of Education, Culture, Sports, Science and Technology, https://gsdm.u-tokyo.ac.jp/en/ (K.N.), "The funders had no role in study design, data collection and analysis, decision to publish, or preparation of the manuscript." JSPS KAKENHI Grant-in-Aid for JSPS Research Fellow (JP20J21472), https://kaken.nii.ac.jp/en/grant/KAKENHI-PROJECT-20J21472/ (K.N.), "The funders had no role in study design, data collection and analysis, decision to publish, or preparation of the manuscript." PRIME from AMED under grant number JP20gm5910023, https://www.amed.go.jp/en/index.html (T.S.), "The funders had no role in study design, data collection and analysis, decision to publish, or preparation of the manuscript." CREST from AMED under grant number JP20gm1210010, https://www.amed.go.jp/en/index.html (T.S.),"The funders had no role in study design, data collection and analysis, decision to publish, or preparation of the manuscript." a Grant-in-Aid for Scientific Research on Innovative Areas (Dynamic regulation of brain function by the Scrap & Build system) (19H04765), https://kaken.nii.ac.jp/en/grant/KAKENHI-PUBLICLY-19H04765/ and (Inflammation Cellular Sociology) (20H04957), https://kaken.nii.ac.jp/en/grant/KAKENHI-PUBLICLY-20H04957/ from the Ministry of Education, Culture, Sports, Science and Technology of Japan (MEXT) (T.S.), "The funders had no role in study design, data collection and analysis, decision to publish, or preparation of the manuscript." JSPS KAKENHI Grants-in-Aid for Young Scientists (17H05096),https://kaken.nii.ac.jp/en/grant/KAKENHI-PROJECT-17H05096/ (T.S.), (18K14831) https://kaken.nii.ac.jp/en/grant/KAKENHI-PROJECT-18K14831/ (S.S.) and (17K15204) https://kaken.nii.ac.jp/en/grant/KAKENHI-PROJECT-17K15204/ (J.T.), "The funders had no role in study design, data collection and analysis, decision to publish, or preparation of the manuscript." Toray Science and Technology Grant, https://www.toray-sf.or.jp/en/activity/grant.html (T.S.),"The funders had no role in study design, data collection and analysis, decision to publish, or preparation of the manuscript." Takeda Science Foundation, https://www.takeda-sci.or.jp/index.html (T.S., S.S., J.T.),"The funders had no role in study design, data collection and analysis, decision to publish, or preparation of the manuscript." Mitsubishi Foundation, https://www.mitsubishi-zaidan.jp/en/ (T.S.),"The funders had no role in study design, data collection and analysis, decision to publish, or preparation of the manuscript." SENSHIN Medical Research Foundation, https://www.smrf.or.jp/ (T.S.),"The funders had no role in study design, data collection and analysis, decision to publish, or preparation of

extracellular space from damaged tissue [4]. These endogenous alarm molecules trigger sterile inflammation and are known as damage-associated molecular patterns (DAMPs). Since the brain is generally a sterile organ, cerebral inflammation is thought to be induced by DAMPs released from damaged brain cells [5]. Recently, the prevention of excess inflammation has been reported to improve the symptoms of neurological diseases and psychiatric disorders [6]. Controlling cerebral inflammation by targeting DAMPs could be a promising therapeutic method for cerebral pathologies.

Ischemic stroke, which is a major cause of death and disability all over the world, is the sudden onset of neurological deficits due to necrosis of brain tissue caused by a severe loss of cerebral blood flow (CBF). The presence of large quantities of DAMPs generated by necrotic brain tissue triggers severe inflammation, which worsens the neurological deficits associated with stroke [7]. DAMPs are thus key molecules in determining an individual's functional prognosis after ischemic stroke. Among the various PRRs that recognize DAMPs, Toll-like receptor 2 (TLR2), and TLR4 are pivotal in cerebral post-ischemic inflammation, given that TLR2 and TLR4 deficiencies considerably decrease the expression of various inflammatory mediators in the ischemic brain [8]. In ischemic stroke, high mobility group box 1 (HMGB1) and the peroxiredoxins (PRXs) have been identified as the DAMPs [8,9]. HMGB1 is a DAMP with an immediate effect, as it is extracellularly released within 6 h after stroke onset and promotes disruption of the blood–brain barrier [9]. The PRXs, on the other hand, trigger the expression of various inflammatory cytokines including IL-23 in infiltrating macrophages through the activation of TLR2 and TLR4 [8]. IL-23 from infiltrating macrophages induces the expression of IL-17 in γδT lymphocytes, which promotes post-ischemic inflammation in the delayed phase of ischemic stroke [10,11]. PRXs identified from brain lysates have been shown to induce the expression of inflammatory cytokines in cultured myeloid cells [8]. This induction is partially decreased by the depletion of PRX proteins in brain lysates.

In this study, we searched for previously unknown DAMPs in brain homogenates. We identified DJ-1 (alternatively known as PARK7) as a major DAMP with a unique peptide sequence that activates TLRs.

## Results

### DJ-1 induces the production of inflammatory cytokines through TLR2 and TLR4 activation

We previously found prominent DAMP activity in the 15 to 25 kDa fractions of brain homogenates [8]. Among candidate DAMPs detected in these fractions by mass spectrometry, we generated the recombinant proteins and added these individually to a culture of bone marrow–derived macrophages (BMMs) to examine DAMP activity. We discovered that a recombinant DJ-1 protein induced mRNA expression of inflammatory cytokines such as TNFα, IL-1β, and IL-23 (a heterodimer of IL-23p19 and IL-12p40) in a dose-dependent manner (**Fig 1A**), while a control protein, GST (glutathione-S-transferase), did not. The further addition of brain homogenates with a recombinant DJ-1 protein did not markedly affect their DAMP activities in bone marrow–derived dendritic cells (BMDCs), indicating little implication of other cerebral DAMPs in DJ-1's DAMP activity (**S1 Fig**). mRNA expression of TNFα, IL-1β, and IL-23 increased within 6 h after the addition of DJ-1 protein in BMMs (**Fig 1B**). Amounts of TNFα and IL-23 protein in the culture supernatant of BMMs increased 3 h after the stimulation by DJ-1, which were significantly higher than GST-treated BMMs (**Fig 1C**).

Since TLRs are major PRRs which mediate the inflammatory responses of various DAMPs, we next explored whether this is also the case for DJ-1 by HEK293 cells expressing each TLR family member. DJ-1-induced activation of nuclear factor kappa B (NF-κB) was investigated

the manuscript." MSD Life Science Foundation, https://www.msd-life-science-foundation.or.jp/ (T. S.),"The funders had no role in study design, data collection and analysis, decision to publish, or preparation of the manuscript." Senri Life Science Foundation, http://www.senri-life.or.jp/ (T.S.),"The funders had no role in study design, data collection and analysis, decision to publish, or preparation of the manuscript." Ono Medical Research Foundation, https://www.ono.co.jp/jp/zaidan/ (T. S.)"The funders had no role in study design, data collection and analysis, decision to publish, or preparation of the manuscript."

**Competing interests:** The authors have declared that no competing interests exist.

**Abbreviations:** ANOVA, analysis of variance; BMDC, bone marrow–derived dendritic cell; BMM, bone marrow–derived macrophage; CBB, Coomassie brilliant blue; CBF, cerebral blood flow; cDNA, complementary DNA; Cys-106, cysteine 106; DAMP, damage-associated molecular pattern; ELISA, enzyme-linked immunosorbent assay; FACS, fluorescence-activated cell sorting; GM-CSF, granulocyte macrophage colony-stimulating factor; GST, glutathione-S-transferase; HMGB1, high mobility group box 1; HRP, horseradish peroxidase; IgG, immunoglobulin G; KO, knockout; LC–MS, liquid chromatography–mass spectrometry; LPS, lipopolysaccharide; MCAO, middle cerebral artery occlusion; M-CSF, macrophage colony-stimulating factor; NCBI, National Center for Biotechnology Information; NF-κB, nuclear factor kappa B; NHS, N-hydroxysuccinimide; OGD, oxygen-glucose deprivation; PBS, phosphate-buffered saline; PCR, polymerase chain reaction; PDB, Protein Data Bank; PGN, peptidoglycan; PRR, pattern recognition receptor; PRX, peroxiredoxin; ROS, reactive oxygen species; TLR, Toll-like receptor; TUNEL, terminal deoxynucleotidyl transferase–mediated dUTP-biotin in situ nick end labeling; WT, wild-type.

using a luciferase reporter assay in these HEK293 cells. Recombinant DJ-1 protein induced luciferase activity in TLR2-expressing HEK293 cells but not in control HEK293 cells (**Fig 2A**). A positive control, the TLR2 ligand peptidoglycan (PGN), also induced luciferase activity in TLR2-expressing HEK293 cells. A similar increase in luciferase activity was also observed by the addition of recombinant DJ-1 protein to TLR4/MD-2/CD14-expressing HEK293 cells, comparable to the increase induced by stimulation with the TLR4 ligand lipopolysaccharide (LPS) (**Fig 2B**). Control GST protein did not induce luciferase activity in any of these cell types. Although we also examined the activation of endosomal TLRs (TLR3, TLR7, or TLR9) by recombinant DJ-1 or intracellular overexpression of DJ-1, we could not detect any lucifer-ase activity in these endosomal-TLRs-expressing HEK293 cells (**Figs 2C and S2**). Therefore, we identified TLR2 and TLR4 as PRRs for DJ-1, and this was confirmed by using myeloid cells generated from the bone marrow of TLR2 or TLR4 single knockout (KO) mice and TLR2/4 DKO mice. As shown in **Figs 2D** and **2E**, the recombinant-DJ-1-induced expression of inflam-matory cytokines was decreased by TLR2 or TLR4 deficiency and almost completely abolished by the deficiency of both TLR2 and TLR4. Thus, TLR2 and TLR4 are PRRs for the DJ-1 protein to trigger inflammatory responses.

## The αG and αH helix region of DJ-1 is important for DAMP activity

Since DJ-1 is known to be an antioxidant protein whose cysteine 106 (Cys-106) is important for reducing reactive oxygen species (ROS) [12], we generated a mutant DJ-1 protein which lacked the ability to form the oxidized state, or dithiothreitol-treated DJ-1 protein. However, these modifications did not affect DAMP activity (**S3 Fig**), suggesting that DAMP activity and antioxidant activity are separable functions of DJ-1. Next, in order to identify the peptide sequence in DJ-1 required for activation of TLR2 and TLR4, we generated various DJ-1 dele-tion mutant peptides fused with GST (**Fig 3A**). We examined IL-23-inducing activities of car-boxyl-terminal deletion mutants using BMMs and found that a deletion between residues 160 and 189 of DJ-1 significantly decreased IL-23-inducing activity (**Fig 3B**). Thus, we focused on the carboxyl terminus region of DJ-1. GST fused to residues 100 to 189 of DJ-1-induced IL-23 to a level similar to full-length DJ-1 protein (**Fig 3C**). While residues 100 to 160 did not induce IL-23 expression in BMMs, induction by residues 160 to 189 was again similar to that of full-length DJ-1 protein (**Fig 3C**). This region is located at the surface of the DJ-1 molecule and contains the αG helix and αH helix regions (**Fig 3D**). Although DJ-1 is an atypical PRX-like peroxidase, the peptide sequence between residues 100 and 120 of DJ-1, which exhibits no DAMP activity, is most similar to the peptide sequence in the β4 sheet and α3 helix regions of PRXs, which are important for the DAMP activities exhibited by the PRXs [8] (**Fig 3E**). As we found no similarity between the peptide sequence of the αG–αH helix region of DJ-1 and that of any other proteins bearing DAMP activity (HMGB1, PRXs, S100A8/A9, etc.), DJ-1 seems to have a unique peptide sequence that triggers the production of inflammatory cytokines.

## DJ-1 is extracellularly released from necrotic brain cells and stimulates infiltrating macrophages

Next, we examined the function of extracellular DJ-1 as a DAMP in the ischemic brain using a murine model of brain ischemia–reperfusion. In the normal brain, DJ-1 protein could be detected in neurons, astrocytes, oligodendrocytes, but not in myeloid cells using tyramide-mediated signal amplification (**S4A Fig**). DJ-1 was not detected in normal, nonischemic neu-rons using nonenhanced staining in the absence of tyramide but was detected in ischemic neu-rons 6 to 12 h after stroke onset (**Fig 4A**), revealing that ischemic stress induces DJ-1 expression in neurons. Twenty-four hours after stroke onset, NeuN-positive cells did not

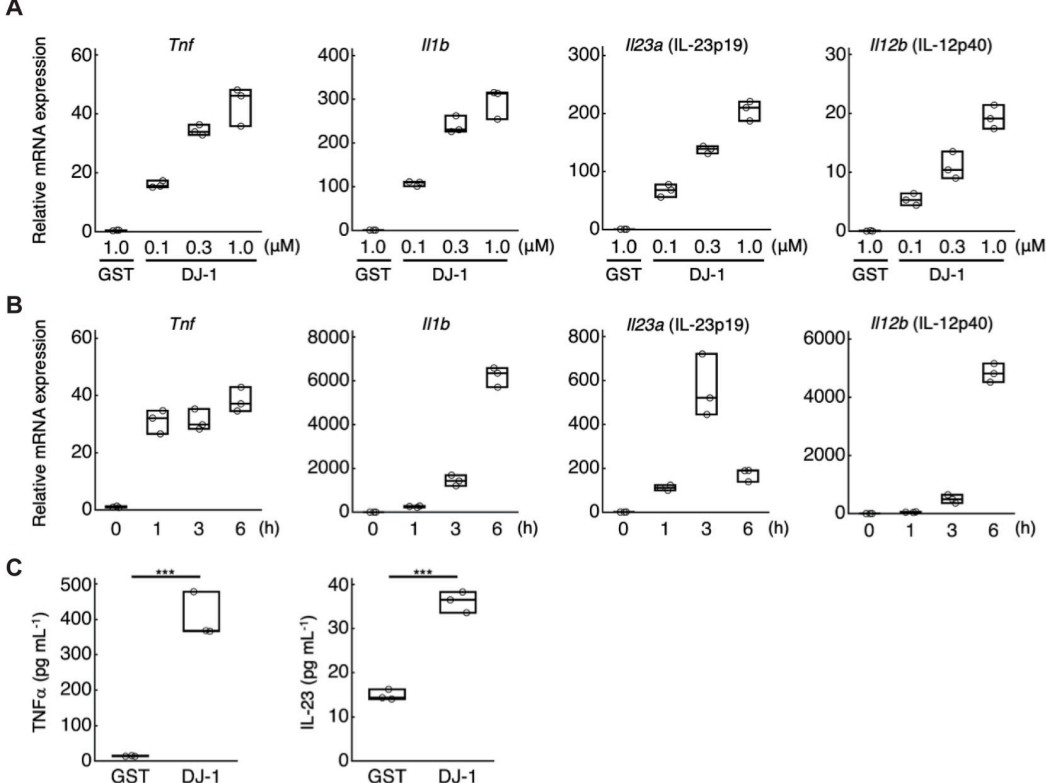

**Fig 1. DJ-1 induces the expression of inflammatory cytokines in myeloid cells. (A)** The mRNA expression levels of inflammatory cytokines in BMMs treated with the indicated concentrations of recombinant DJ-1 or GST protein for 1 h. Each value indicates the relative mRNA expression level compared to untreated BMM. **(B)** Time-dependent changes in mRNA expression level of inflammatory cytokines in BMMs treated with 1 μM of recombinant DJ-1 protein. **(C)** TNFα and IL-23 protein expression levels in the BMM supernatant 3 h after treatment with 1 μM of recombinant DJ-1 or GST protein. $^{***}p < 0.001$ vs. GST-treated BMMs **(C)** (two-sided Student $t$ test [**C**]). All experiments were performed in triplicate. The results are representative of 2 **(A, B)** or 3 **(C)** independent experiments. The data underlying this figure can be found in S1 Data. BMM, bone marrow–derived macrophage; GST, glutathione-S-transferase.

survive in the infarct area and DJ-1 was detected in debris-like structures whereas staining with control immunoglobulin G (IgG) did not yield any signals in ischemic brains (**Fig 4A**). This extracellular DJ-1-including debris was also observed even when the ischemic brain tissue was not reperfused, indicating that the reperfusion after brain ischemia is not necessary for the extracellular release of DJ-1 (**S5A and S5B Fig**). The specificity of our anti-DJ-1 antibody was confirmed by using brain tissue of DJ-1 KO mice (**S4B and S4C Fig**). Debris containing DJ-1 was observed scattered around terminal deoxynucleotidyl transferase–mediated dUTP-biotin in situ nick end labeling (TUNEL)-positive necrotic brain cells in the infarct region 24 h after stroke onset (**Fig 4B**). These results suggested that ischemic stress induced the DJ-1 expression within neurons as intracellular DJ-1 had antioxidative effects, but this intracellular DJ-1 was passively released into the extracellular space from dead neurons when ischemic stress resulted in neuronal cell death. Indeed, the primary cultured neurons subjected to oxygen-glucose deprivation (OGD) expressed DJ-1 intracellularly, while the amount of extracellular DJ-1 in the culture supernatant increased along with the decrease of survived neurons enough to activate BMDCs in vitro (**S6A and S6B Fig**).

We then compared the extracellular release of DJ-1 between the peri-infarct area and the infarct area. Around the infarct boundary zone, DJ-1 expression was majorly observed within NeuN-positive cells in the peri-infarct area even 24 h after stroke onset (**Fig 4C**); however, the

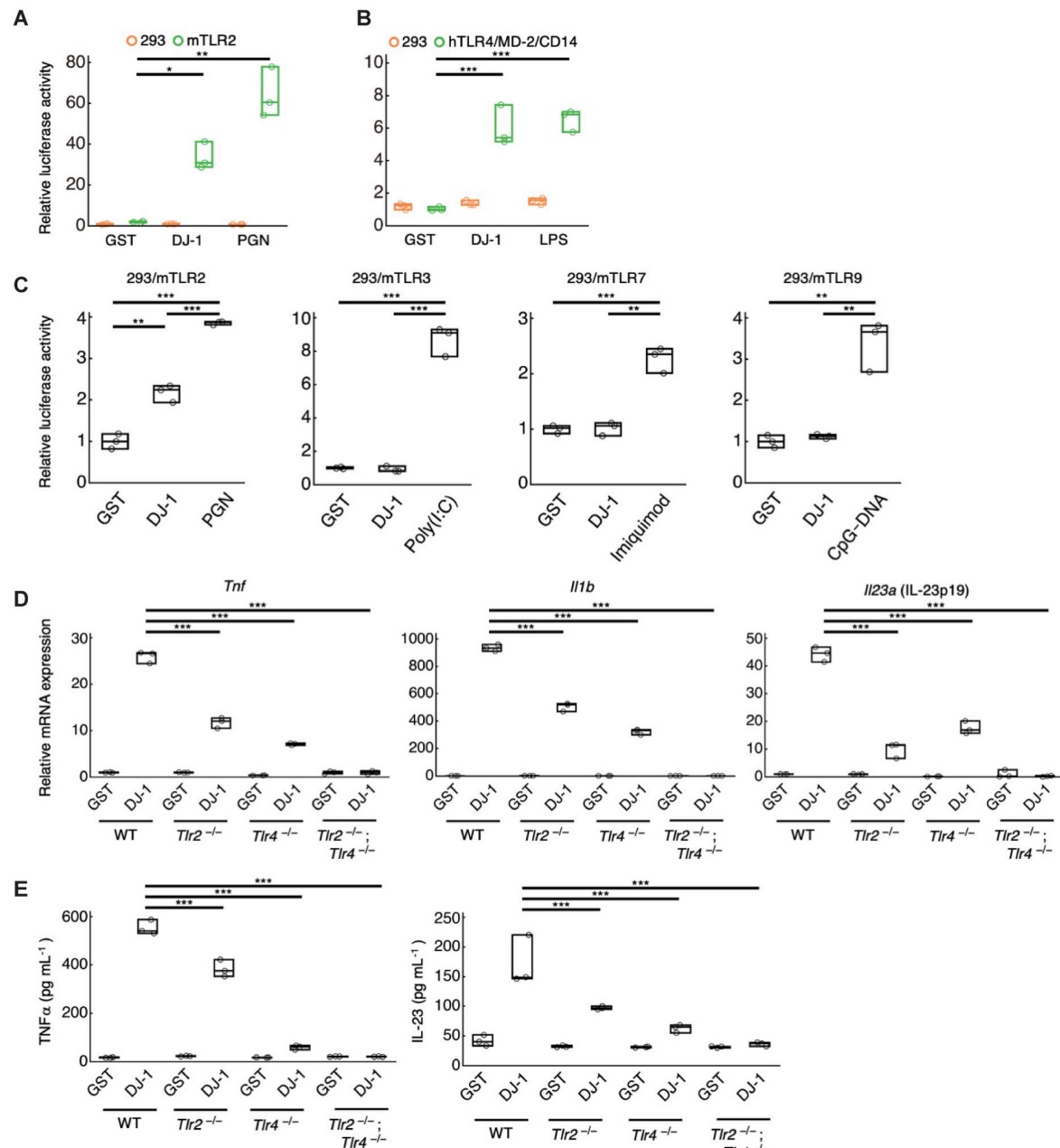

**Fig 2. DJ-1 activates TLR2 and TLR4 to trigger inflammation. (A)** The relative luciferase activity of NF-κB reporter in HEK293 cells or murine stably TLR2-expressing HEK293 cells treated with 1 μM of recombinant DJ-1 protein or PGN (TLR2 ligand). **(B)** The relative luciferase activity of NF-κB reporter in HEK293 cells or human stably TLR4/MD-2/CD14-expressing HEK293 cells treated with 1 μM of recombinant DJ-1 protein or LPS (TLR4 ligand). The relative values compared to those in HEK293 cells treated with 1 μM of control GST protein are shown **(A, B)**. **(C)** Relative luciferase activity of the NF-κB reporter in each murine transiently TLRs-expressing HEK293 cells treated with 1 μM of recombinant GST (as a control) or DJ-1 protein or TLR ligands (Poly(I:C): a TLR3 ligand, Imiquimod: a TLR7 ligand, and CpG-DNA: a TLR9 ligand). **(D)** The mRNA expression levels of inflammatory cytokines in WT, TLR2-deficient, TLR4-deficient, TLR2- and TLR4-double-deficient BMMs 1 h after stimulation with 0.3 μM of recombinant DJ-1 protein or control GST protein. The relative values compared to the mRNA expression levels in GST-treated BMMs are shown. **(E)** The protein expression levels of inflammatory cytokines in the WT, TLR2-deficient, TLR4-deficient, TLR2- and TLR4-double-deficient BMMs (for TNFα), and BMDCs (for IL-23) 3 h after treatment with 1 μM of recombinant DJ-1 or control GST protein $^*p < 0.05$, $^{**}p < 0.01$, $^{***}p < 0.001$ (one-way ANOVA with Dunnett correction [**A–E**]). All experiments were performed in triplicate. The results are representative of 2 (**A, E**) or 3 (**B–D**) independent experiments. The data underlying this figure can be found in S1 Data. ANOVA, analysis of variance; BMDC, bone marrow–derived dendritic cell; BMM, bone marrow–derived macrophage; GST, glutathione-S-transferase; LPS, lipopolysaccharide; NF-κB, nuclear factor kappa B; PGN, peptidoglycan; TLR, Toll-like receptor; WT, wild-type.

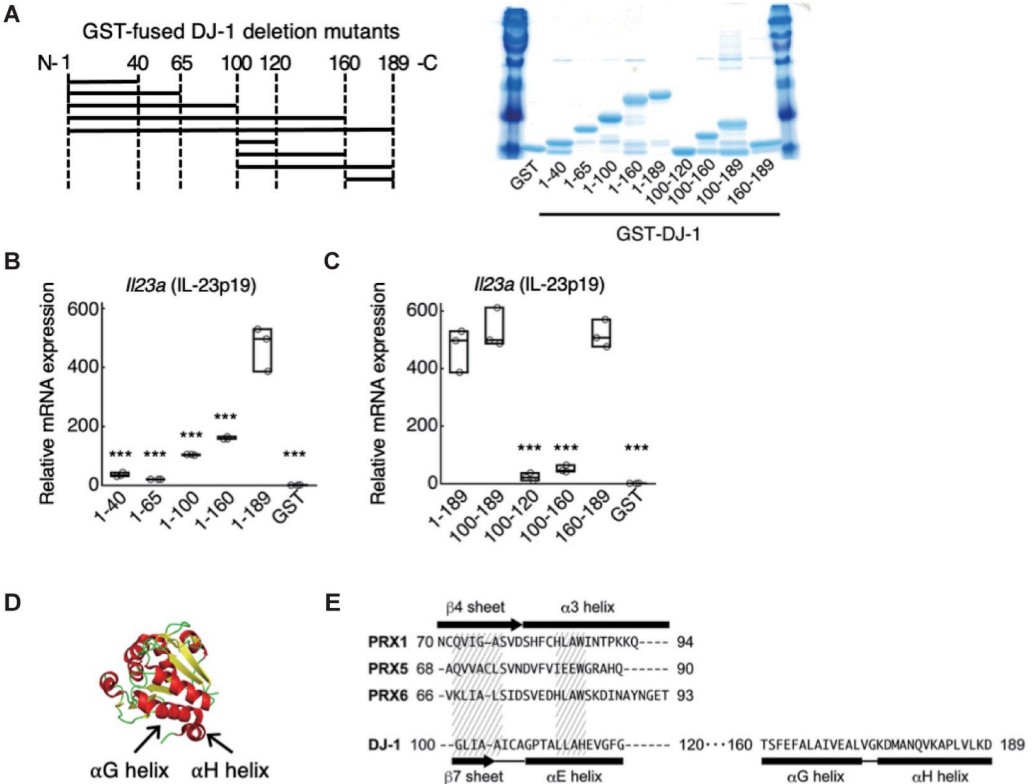

**Fig 3. The αG and αH helix region of DJ-1 is essential for DAMP activity. (A)** Diagram of generated deletion mutant peptides of DJ-1 fused with GST (left panel). The number of amino acid residues contained in each GST-fused DJ-1 peptide is shown on the x-axis. The amount of generated deletion mutant peptides was examined by SDS-PAGE with CBB staining (right panel). **(B)** IL-23p19-inducing activities of carboxyl-terminal deletion mutants of DJ-1. **(C)** IL-23p19-inducing activities of GST-fused peptides containing the carboxyl-terminal region of DJ-1. **(D)** αG helix (DJ-1$_{160-}$ $_{173}$: peptides between residues 160 and 173 of DJ-1) and αH helix (DJ-1$_{175-189}$) are indicated in the crystal structure of DJ-1 protein (PDB ID: 1P5F). **(E)** Comparison of peptide sequences among PRX1$_{70-94}$, PRX5$_{68-90}$, and PRX6$_{66-93}$ containing the β4 sheet and α3 helix of PRXs known to be important for DAMP activity, DJ-1$_{100-120}$ containing the β7 sheet and αE helix, and DJ-1$_{160-189}$ containing the αG and αH helix. Similarities among the amino acid residues of PRXs and DJ-1 are indicated by hatched areas. $^{***}p < 0.001$ vs. BMMs treated with DJ-1$_{1-189}$ **(B, C)** (one-way ANOVA with Dunnett correction [**B, C**]). Experiments were performed in triplicate, and the results are representative of 2 independent experiments **(B, C)**. The data underlying this figure can be found in S1 Data. ANOVA, analysis of variance; BMM, bone marrow–derived macrophage; CBB, Coomassie brilliant blue; DAMP, damage-associated molecular pattern; GST, glutathione-S-transferase; PRX, peroxiredoxin.

extracellular DJ-1-including debris that was observed separate from cellular membranes (stained with pan-cadherin antibody) was detected not only in the peri-infarct area but also markedly in the infarct area (**Fig 4D**). The extracellular release of DJ-1 coincided with the increase of myeloid cell infiltration in the ischemic brain (**Figs 4A** and **S7**), and the extracellular DJ-1-including debris was in direct contact with the F4/80-positive cellular membranes of infiltrating myeloid cells in the infarct region (**Fig 4E**). We also measured the contact area per cell and found that it was similar between the peri-infarct area and the infarct area (**Fig 4F**), suggesting the comparable stimulation of infiltrating myeloid cells by extracellular DJ-1 between the peri-infarct area and infarct area. The protein–protein interaction of DJ-1 with TLR2 or TLR4 in infiltrating myeloid cells was successfully detected in the ischemic brain on day 1 after stroke onset (**Fig 4G and 4H**). These observations revealed that DJ-1 was released into the extracellular space from necrotic neurons where it directly activated TLR2 and TLR4 in infiltrating myeloid cells.

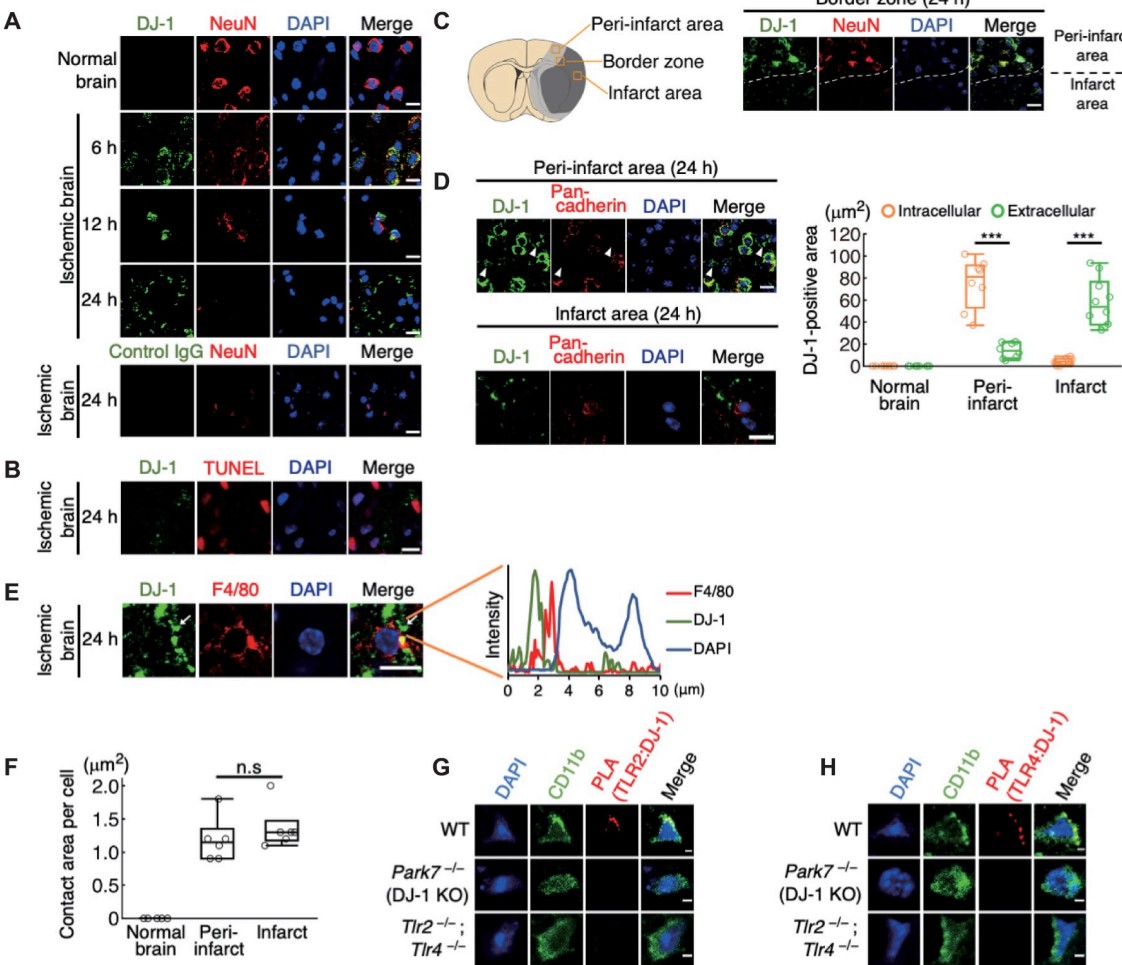

**Fig 4. DJ-1 is extracellularly released from necrotic neurons and contacts infiltrating myeloid cells in the ischemic brain.** (A) Time-dependent changes of immunohistochemistry in the brain tissue before and after ischemic stroke. Control IgG was used as a control. (B) Immunohistochemical staining of DJ-1 around the TUNEL-positive dead brain cells. (C) Left panel: schematic diagram of the peri-infarct area (light gray region) and infarct area (dark gray region). The squares indicate the location where each image shown in **Fig 4C** or **Fig 4D** was captured. Right panel: immunohistochemistry in the border zone between peri-infarct area and infarct area 24 h after stroke onset. The dashed white line indicates the infarct border. (D) Immunohistochemistry in the peri-infarct area and infarct area 24 h after stroke onset (left panel). Arrowhead indicates the extracellular DJ-1-including debris. Right panel: quantification of intracellular or extracellular DJ-1-positive areas in the indicated region ($n$ = 10 mice for normal brain and infarct area, $n$ = 8 mice for peri-infarct area) ***$p$ < 0.001 vs. intracellular DJ-1-positive area (E) Immunohistochemistry in the infarct region 24 h after stroke onset. The white arrow indicates the direct contact of DJ-1-including debris with the cellular membranes of infiltrating myeloid cells. Fluorescence intensity along the white arrow is shown in the right panel. (F) Quantification of areas where DJ-1-including debris contacted the cellular membranes in each infiltrating myeloid cells ($n$ = 6 mice for each group). n.s., not significant vs. peri-infarct area. (G, H) The image of the PLA to detect the protein–protein interaction between DJ-1 and TLR2 (G) or TLR4 (H) in the infarct area of each mouse 24 h after stroke onset. All images were captured by confocal laser microscopy (one-way ANOVA with Dunnett correction [**D, F**]). (scale bars: 10 μm [**A–E**], 2 μm [**G, H**]). The data underlying this figure can be found in S1 Data. ANOVA, analysis of variance; IgG, immunoglobulin G; PLA, proximity ligation assay; TUNEL, terminal deoxynucleotidyl transferase–mediated dUTP-biotin in situ nick end labeling.

## DJ-1 functions as a DAMP in ischemic stroke

To prove our hypothesis that the activation of infiltrating myeloid cells by DJ-1 leads to the progression of ischemic neuronal injury, we examined whether the deficiency of DJ-1 exerted an immunosuppressive effect in the murine model of ischemic stroke. We found that DJ-1 deficiency significantly decreased the expression of inflammatory cytokines such as TNFα, IL-

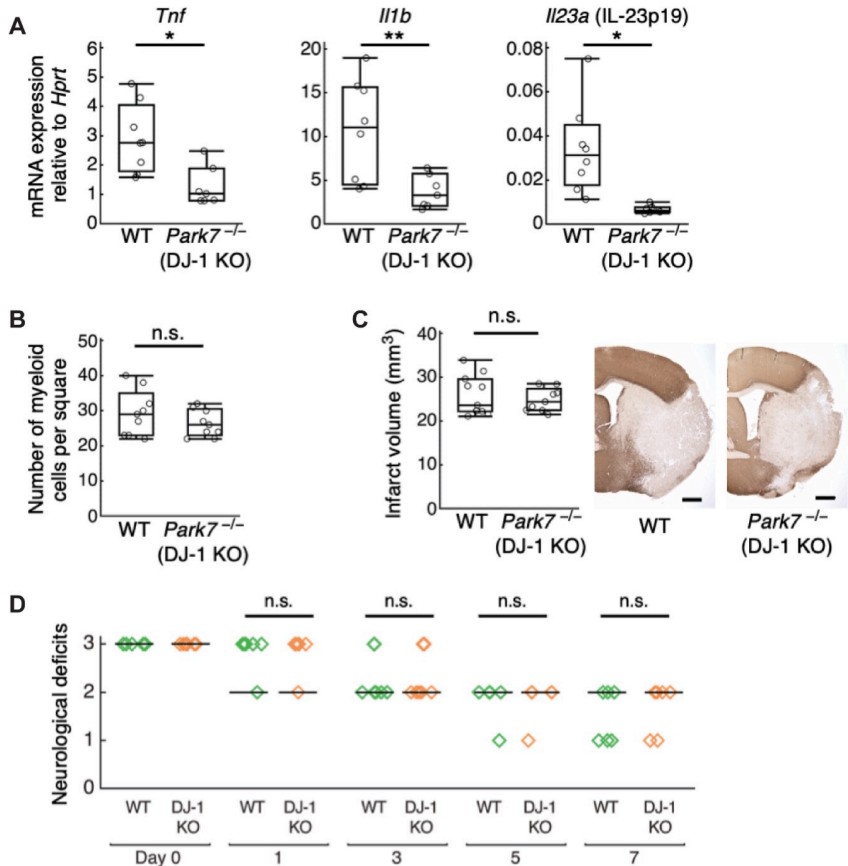

**Fig 5. DJ-1 deficiency decreased the inflammatory cytokine expression but not attenuated ischemic brain injury.**
**(A)** The mRNA expression levels of inflammatory cytokines in infiltrating immune cell–enriched cellular population collected from day 1 post-ischemic brains of WT or DJ-1 KO mice ($n = 8$ mice for WT, $n = 7$ mice for DJ-1 KO). The relative values compared to *Hprt* are shown. $^*p < 0.05$, $^{**}p < 0.01$ vs. WT mice. **(B)** The number of infiltrating CD11b$^+$ myeloid cells detected by immunohistochemistry in the infarct area 24 h after stroke onset ($n = 9$ mice for each group, per 0.16 mm$^2$). **(C)** Infarct volume of WT or DJ-1-deficient mice on day 7 after stroke onset ($n = 12$ mice for WT, $n = 9$ mice for DJ-1 KO) n.s., not significant vs. WT. **(D)** Neurological deficits of WT or DJ-1-deficient mice until day 7 after stroke onset ($n = 6$ mice for WT, $n = 6$ mice for DJ-1 KO). (Two-sided Student *t* test [**A–C**], Wilcoxon rank sum test with Bonferroni correction [**D**]). The data underlying this figure can be found in S1 Data. KO, knockout; WT, wild-type.

1β, and IL-23p19 in the population enriched with infiltrating immune cells collected from ischemic brain (**Fig 5A**). The number of infiltrating macrophages/neutrophils was not altered by DJ-1 deficiency (**Fig 5B**), revealing the important role of DJ-1 in triggering the inflammatory cytokine production from infiltrating macrophages/neutrophils. Despite the significant attenuation of inflammatory cytokine expression by DJ-1 deficiency, the infarct volume and neurological deficits were similar between wild-type (WT) and DJ-1 KO mice (**Fig 5C and 5D**). There was no significant difference in CBF and survival rate between WT and DJ-1 KO mice (**S1A Table**). Since these results were explained by the cancellation of the inflammatogenic (neurotoxic) effect of extracellular DJ-1 and the antioxidative (neuroprotective) effect of intracellular DJ-1 that was previously reported [13], we next examined the administration of a DJ-1-specific antibody to neutralize extracellular DJ-1.

Compared to control IgG antibody, the DJ-1-specific antibody significantly decreased mRNA expression of IL-23p19 in the population enriched with infiltrating immune cells collected from ischemic brain 24 h after stroke onset (**Fig 6A**). mRNA expression of TNFα and

 Extracellularly released DJ-1 from necrotic neurons activates immune cells in ischemic brain

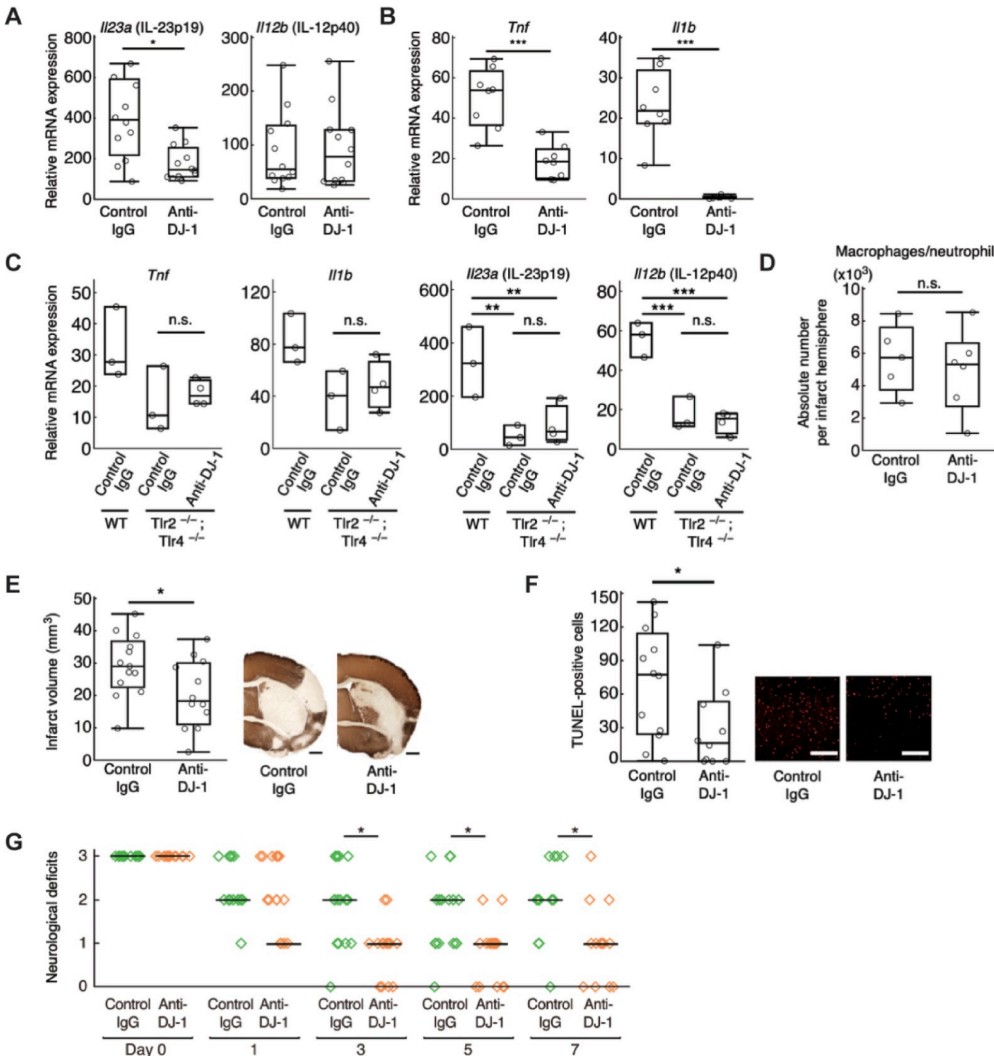

**Fig 6. Administration of anti-DJ-1 antibody decreased the inflammatory cytokine expression and attenuated ischemic neuronal injury.** (A) The relative mRNA expression levels in infiltrating immune cell–enriched cellular population collected from day 1 post-ischemic brains of mice treated with indicated antibody ($n$ = 12 mice for each group), compared to those in sham-operated mice. (B) The relative mRNA expression levels in the day 3 post-ischemic brain tissue ($n$ = 8 mice for each group), compared to those in sham-operated mice. (C) The relative mRNA expression levels in infiltrating immune cell–enriched cellular population collected from day 1 post-ischemic brains of WT or TLR2/4 DKO mice treated with indicated antibody ($n$ = 3 mice for WT and control IgG–administered TLR2/4 DKO mice, $n$ = 4 mice for anti-DJ-1 antibody-administered TLR2/4 DKO mice). n.s., not significant vs. control IgG–administered TLR2/4 DKO mice. (D) The absolute number of CD45[high]CD11b[high] population (macrophages/neutrophils) collected from the day 3 post-ischemic brain ($n$ = 5 mice for control IgG, $n$ = 6 mice for anti-DJ-1 antibody). n.s., not significant vs. control IgG. (E) Infarct volume of mice treated with indicated antibody on day 7 after stroke onset ($n$ = 13 mice for control IgG, $n$ = 12 mice for anti-DJ-1 antibody). (F) The absolute number of TUNEL-positive neuronal cells in the day 7 post-ischemic peri-infarct area of mice treated with indicated antibody ($n$ = 12 mice for control IgG, $n$ = 10 mice for anti-DJ-1 antibody) (scale bars: 500 μm [**E**], 100 μm [**F**]). (G) Neurological deficits of mice treated with each antibody until day 7 after stroke onset. ($n$ = 19 mice for control IgG, $n$ = 18 mice for anti-DJ-1 antibody). n.s., not significant, *$p < 0.05$, ***$p < 0.001$ vs. mice treated with control IgG antibody. (**A, B, D–G**) (two-sided Student $t$ test [**A, B, D–F**]. One-way ANOVA with Dunnett correction [**C**]. Wilcoxon rank sum test with Bonferroni correction [**G**]). The data underlying this figure can be found in S1 Data. ANOVA, analysis of variance; DKO, double knockout; IgG, immunoglobulin G; TUNEL, terminal deoxynucleotidyl transferase–mediated dUTP-biotin in situ nick end labeling; WT, wild-type.

IL-1β in ischemic brain tissue on day 3 after stroke onset was also decreased in mice treated with anti-DJ-1 antibody (**Fig 6B**). The administration of anti-DJ-1 antibody did not decrease the inflammatory cytokine expression in the population enriched with infiltrating immune cells collected from TLR2/4 DKO mice (**Fig 6C**), revealing the important role of TLR2 and TLR4 for recognizing extracellular DJ-1 in infiltrating immune cells. The number of infiltrating myeloid cells was not altered by the administration of antibodies (**Fig 6D**). Therefore, the neutralization of extracellular DJ-1 did not inhibit the infiltration of immune cells but suppressed the neurotoxic inflammation in the ischemic brain tissue. Glial cells such as microglia or astrocytes did not play a major role in the production of inflammatory cytokines triggered by extracellular DJ-1 (**S8 Fig**). This was consistent with the fact that the administration of anti-DJ-1 antibody attenuated neither the cerebral post-ischemic inflammation nor ischemic neuronal damage if macrophages were depleted by clodronate liposome administration (**S9 Fig**). Furthermore, the extracellular DJ-1 did not directly induce the cell death in primary cultured neurons but triggered the production of neurotoxic inflammatory mediators through the activation of macrophages in vitro (**S10A and S10B Fig**).

Finally, we examined the neuroprotective effect of anti-DJ-1 antibody against ischemic stroke. The administration of anti-DJ-1 antibody immediately after stroke onset significantly reduced the infarct volume compared to the administration of control IgG antibody (**Fig 6E**). There was no significant difference in CBF between the 2 groups, although a slight improvement in the survival rate was seen in the mice treated with anti-DJ-1 antibody (**S1B Table**). The number of TUNEL-positive dead neuronal cells in the peri-infarct area on day 7 after stroke onset was significantly decreased by the administration of anti-DJ-1 antibody (**Fig 6F**). Consistent with this, the neurological deficits in the mice treated with anti-DJ-1 antibody were significantly improved on day 7 after stroke onset (**Fig 6G**). Altogether, our results indicate that DJ-1 is a DAMP molecule that promotes neuronal injury after ischemic stroke.

## Discussion

DJ-1 was first identified as an oncogenic protein and later recognized as a pivotal protein associated with the onset of familial Parkinson disease [14,15]. DJ-1-deficient mice are less effective at scavenging mitochondrial hydrogen peroxide in old age, indicating that DJ-1 is an atypical PRX-like peroxidase that catalyzes ROS [16]. Increased expression levels of DJ-1 within brain cells are observed in chronic neurodegenerative diseases [13]. Thus, DJ-1 is believed to prevent neurodegeneration by reducing oxidative stress within brain cells. A neuroprotective role of intracellular DJ-1 against ischemic brain injury has been reported by using an endothelin-1 injection model [13]. In contrast, we did not observe significant differences in infarct volume between WT and DJ-1 KO mice (**Fig 5C**). This discrepancy is likely due to the different experimental methods used for the induction of brain ischemia. We used a middle cerebral artery occlusion (MCAO) model. A drastic infiltration of immune cells is generally observed in MCAO models. Thus, the neurotoxic effect of inflammation induced by extracellular DJ-1 should be greater compared to that in an endothelin-1 injection model. In our MCAO model, the neuroprotective effect of intracellular DJ-1 seems to be canceled by the inflammation-mediated neuronal damage triggered by extracellular DJ-1.

Our current study reveals that extracellularly released DJ-1 protein from necrotic brain cells triggers cerebral post-ischemic inflammation through the activation of TLR2 and TLR4 in infiltrating myeloid cells. We observed an increased expression of DJ-1 within neurons in the ischemic brain, suggesting that intracellularly accumulated DJ-1 is released into the extracellular space where it then functions as a DAMP. Thus, DJ-1 seems to have 2 opposing functions in the intracellular and extracellular contexts (**Fig 7**). Since important roles of DJ-1 have

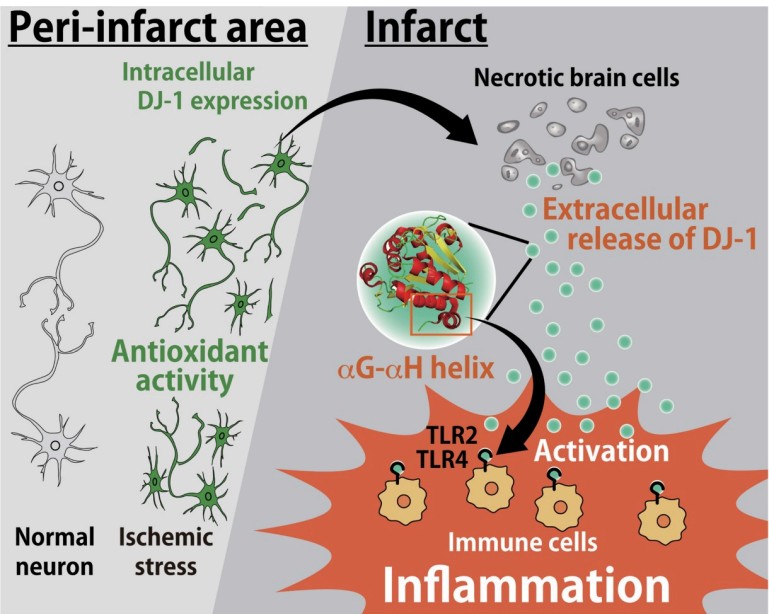

**Fig 7. Schematic model of the roles of DJ-1 in the ischemic brain injury.** In the peri-infarct area, the expression level of DJ-1 within the neurons increases due to ischemic stress. Intracellular DJ-1 plays a neuroprotective role by catalyzing ROS whereas if ischemic stress results in neuronal cell death, accumulated DJ-1 within ischemic neurons is passively released into the extracellular space. The αG–αH helix of extracellular DJ-1 directly activates TLR2 and TLR4 in infiltrating immune cells to trigger sterile inflammation. Therefore, DJ-1 has 2 opposing functions: extracellularly (inflammatory molecule as a DAMP) and intracellularly (antioxidant activity) in the pathology of ischemic stroke. DAMP, damage-associated molecular pattern; ROS, reactive oxygen species.

been demonstrated in several neurodegenerative diseases [14,17–20], the function of extracellular DJ-1 as an inflammatory DAMP may be implicated in the pathophysiology of neurodegenerative diseases including Parkinson disease.

DJ-1 protein is an atypical PRX-like peroxidase; therefore, we compared the crystal structures and amino acid sequences between DJ-1 and PRX family proteins. Although PRXs have 1 or 2 cysteine residues which are conserved among PRX family proteins and are important for their antioxidant activities, DJ-1 also has 2 cysteine residues (Cys-53 and Cys-106) in humans and 3 cysteine residues (Cys-53, Cys-106, and Cys-121) in mice. Among these, the oxidation-sensitive Cys-106 of DJ-1 has been demonstrated to be essential for reducing oxidative stress within neuronal cells after ischemic stroke [12,13]. Hydrogen bonding between glutamate-18 (Glu-18) and Cys-106 of DJ-1 catalyzes ROS by converting between their reduced and oxidized forms [21]. Once DJ-1 is released into the extracellular space, however, its antioxidant activity is abolished as its cysteine residues are rapidly oxidized [22]. Crystal structure analysis reveals almost no differences between reduced and oxidized DJ-1, suggesting that the antioxidative function of DJ-1 does not significantly affect its DAMP activity [21]. We identified the αG–αH helix region of DJ-1 as an active center of DAMP activity that has a unique structure compared to PRX family proteins. The neutralization of PRX1, PRX2, PRX5, and PRX6 is necessary for the suppression of inflammatory cytokine production in ischemic stroke [8], while the significant attenuation of inflammation was observed even by the neutralization of only DJ-1. This may be advantageous for developing the therapeutic method of ischemic stroke.

TLR2 and TLR4 are pivotal PRRs in cerebral sterile inflammation. DJ-1 can activate both TLR2 and TLR4, although the adaptor proteins MD2 and CD14 are necessary for TLR4

activation. The peptide sequence of DJ-1, especially in the αG–αH helix of DJ-1, is highly conserved between bacteria and mammals, suggesting that this conserved peptide sequence is commonly recognized by TLRs. A previous report has demonstrated that the αG–αH helix is necessary for the dimerization of the DJ-1 protein, which may enable its antioxidant activity [23]. However, dimerization of the DJ-1 protein seems unnecessary for the activation of TLR2 and TLR4, since even the GST-fusion αG–αH helix peptide induces almost the same level of DAMP activity as the full-length DJ-1 does. Furthermore, the timing of extracellular release of DAMPs is also important for the activation of infiltrating immune cells in the damaged tissue. The infiltration of macrophages and neutrophils becomes marked within 24 h after stroke onset, and the extracellular release of DJ-1 in the ischemic brain coincides with this infiltration of immune cells. Among the inflammatory cytokines, the expression of IL-23 in the ischemic brain is especially dependent on the TLR2 and TLR4 signaling pathway. IL-23 not only promotes ischemic neuronal injury on day 1 but also sustains the inflammatory response several days after the onset of ischemic stroke [11]. This is consistent with our current finding that the administration of anti-DJ-1 antibody decreased the expression of IL-23 in infiltrating immune cells on day 1 after stroke onset, leading to the suppression of TNFα and IL-1β expression in ischemic brain tissue on day 3 after stroke onset and to neuroprotection against ischemic brain injury.

We discovered the previously unknown function of extracellular DJ-1 as a DAMP. DJ-1 could thus be a therapeutic target to prevent excessive inflammation and neuronal injury after ischemic stroke. In addition, DJ-1 expression has been reported to increase in cancer cells, and this increase is important for cancer pathology [24,25]. Increased expression of DJ-1 within brain cells has also been observed in several neurodegenerative diseases, such as Parkinson disease, amyotrophic lateral sclerosis, Creutzfeldt–Jakob disease, Huntington disease, and Alzheimer disease [26–30]. Extracellular DJ-1 may affect the pathologies of cancer and neurodegenerative diseases and could also be a therapeutic target against inflammatory diseases and tissue injuries.

## Materials and methods

### Ethics statement

All animal experiments were performed according to the guidelines and approved by the ethics committee and the animal research committee of the Tokyo Metropolitan Institute of Medical Science (approval No. 17001). The animal care and use protocol adhere to the Act on Welfare and Management of Animals of Japan.

### Mice

*DJ-1*-deficient mice (*Park7*$^{-/-}$) were purchased from the Jackson Laboratory [31]. TLR2 and TLR4 double-deficient mice were kindly provided by Professor Shizuo Akira of Osaka University, Japan. All mice were on a C57BL/6 background.

### Middle cerebral artery occlusion model

A mouse model of transient MCAO was induced by means of an intraluminal suture as described previously [8]. Briefly, the mice were anesthetized by using isoflurane (1% to 3% inhaled; Pfizer, USA) in a mixture of oxygen and nitrous oxide. We measured CBF before and during brain ischemia using laser Doppler flowmetry at the ipsilateral parietal bone (1–2 mm posterior to bregma). The resting CBF value of each mouse was regarded as the baseline for its mouse, and changes in CBF after the induction of brain ischemia were calculated as

percentages of the resting value. After ligation of the right common carotid artery (CCA), the right middle cerebral artery (MCA) was occluded with a 7–0 ethilon (Johnson & Johnson, USA) monofilament with a rounded tip. We inserted an 11-mm-long filament from the right CCA. The distance from a suture tip to the right CCA bifurcation was 9 mm. Mice with more than 60% reduction of CBF confirmed by laser Doppler flowmetry were included in the experimental results. Male mice (aged 8 to 10 weeks) were used for all MCAO experiments. Mice with less than 60% reduction of CBF or no neurological deficits immediately after MCAO were excluded from the experiments, but no mice were excluded in this study. Head temperature was kept at 36˚C with a heat lamp during MCAO. Sixty minutes after MCAO, the brain was reperfused by withdrawing the intraluminal suture. Two hundred micrograms of control IgG or anti-DJ-1 antibody were intravenously administered from the retro-orbital venous sinus immediately after the induction of brain ischemia. To measure infarct volume, each phosphate-buffered saline (PBS)-perfused brain was fixed with 4% paraformaldehyde/PBS and embedded in paraffin. A 5-μm section was deparaffinized and stained with anti-MAP2 antibody (Sigma-Aldrich, USA, 1:1000 dilution, Cat# M4403) (the details of this procedure are described elsewhere [11]). The infarct area was defined as the MAP2-negatively stained area. Infarct volume was calculated as (contralateral hemisphere − [ischemic hemisphere − infarct area]). Neurological deficits in mice were examined in a blinded fashion and scored using Bederson's 4-point scale neurological score method (0 = no observable deficit, 1 = forelimb flexion, 2 = decreased resistance to lateral push without circling, 3 = same behavior as grade 2, with circling) as previously described [32]. To investigate the expression of inflammatory cytokines, infiltrating immune cells collected by Percoll-gradient centrifugation (GE Healthcare, USA) or ischemic brain tissues were lysed with RNAiso Plus (Takara, Japan). Purified total RNA was transcribed to complementary DNA (cDNA) using a High-Capacity cDNA Reverse Transcription Kit (Applied Biosystems, USA) with random primers. Quantitative polymerase chain reaction (PCR) was performed using SsoFast EvaGreen Supermix (Bio-Rad, USA) on a CFX96 Real-Time System device (Bio-Rad, USA) with different sets of quantitative PCR primers (S2 Table). To deplete macrophages from mice, 100 μL of clodronate liposomes (Formu-Max Scientific, USA) was intraperitoneally administered 3 h before MCAO.

## Preparation of recombinant proteins

cDNA encoding candidate proteins identified by liquid chromatography–mass spectrometry (LC–MS) analysis (described elsewhere [8]) were cloned from a murine brain cDNA library. cDNA constructs were inserted into the pGEX6P-3 plasmid (GE Healthcare, USA) and expressed as GST-fusion proteins in High-efficiency BL21(DE3) Competent Cell (GMbiolab, Taiwan). GST-fusion proteins were purified using glutathione-Sepharose 4B columns (GE Healthcare, USA). Protein-bound glutathione beads were thoroughly washed with cold PBS 6 times, and thereafter eluted with 20 mM reduced glutathione (recombinant GST or GST-fusion peptides) or incubated with PreScission Protease (GE Healthcare, USA) overnight at 4˚C to remove the GST tag. These recombinant proteins were incubated with Affi-Prep Polymyxin Media (Bio-Rad, USA) for 4 h at 4˚C to remove endotoxins and endotoxin-bound proteins. To generate modified DJ-1 proteins, which lack the ability to form its oxidized state, recombinant DJ-1 protein was treated with 1 mM dithiothreitol and incubated with 55 mM iodoacetamide (Wako Pure Chemical Industries, Japan) in a dark chamber at room temperature for 1 h. We examined the purity of these recombinant proteins by SDS-PAGE with Coomassie brilliant blue (CBB) staining. Recombinant GST protein was used as a negative control for cytokine induction in BMMs.

## Generation of rabbit polyclonal antibody

Rabbits were immunized with recombinant DJ-1 protein. N-hydroxysuccinimide (NHS)-Sepharose beads (GE Healthcare, USA) were crosslinked with recombinant proteins according to the manufacturer's instructions and were used for the purification of antibodies against DJ-1 protein. IgG antibodies were purified from rabbit serum using protein A sepharose beads (GE Healthcare, USA), and these IgG antibodies were administered to brain ischemia model mice as a control antibody. The specificity of the anti-DJ-1 antibody thus generated was confirmed by western blotting analysis of lysates from ischemic brain tissue. Sample brain tissues were weighed, and brain lysate samples were prepared by homogenizing brain tissue with lysis buffer. Twenty micrograms of brain lysate were examined by SDS-PAGE, and proteins were blotted on PVDF membrane (Immobilon, Merck, Germany). Membrane was blocked with 2% skim milk and subsequently incubated with anti-DJ-1 antibody or anti-beta actin antibody (Proteintech, USA, 1:1000 dilution, Cat# 20536-1-AP) overnight. Membrane was washed with PBST and incubated with secondary antibody for 1 h. Designated proteins were detected by Chemi-Lumi One (Nacalai Tesque, Japan). Chemiluminescence signal was detected using a Luminescent Image Analyzer LAS-3000 (Fuji, Japan).

## Primary bone marrow–derived macrophage and dendritic cell culture

The femur and tibia were removed from each mouse, and both ends of each bone were cut off. Bone marrow was forced out into RPMI-1640, and the suspension was filtered with a 40-μm filter. Red blood cells were lysed with 17 mM Tris and 48 mM $NH_4Cl$ and centrifuged for 5 min at room temperature. To generate BMMs or BMDCs, the obtained bone marrow cells were cultured in RPMI-1640 containing 10% FBS with 10 ng mL$^{-1}$ mouse macrophage colony-stimulating factor (M-CSF, Peprotech, USA) to generate BMMs or 10 ng mL$^{-1}$ mouse granulocyte macrophage colony-stimulating factor (GM-CSF, Peprotech, USA) to generate BMDCs in a humidified 5% $CO_2$ atmosphere at 37˚C for 6 days. The cultured BMMs were collected with dissociation buffer (10 mM EDTA/PBS) and were used for subsequent experiments. For experiments with BMMs or BMDCs activated by DAMPs, BMMs or BMDCs were cultured with 1 μM of recombinant GST (control) or 0.1 to 1 μM DJ-1 protein or brain homogenates. The preparation of brain homogenates was described previously [8]. The mRNA expression levels were examined by quantitative PCR. TNFα and IL-23 expression in the culture medium was examined using enzyme-linked immunosorbent assay (ELISA) (Invitrogen, USA for TNFα, Biolegend, USA for IL-23).

## NF-κB luciferase assay

HEK293 cells expressing murine TLR2 or human TLR4 and CD14/MD2 were purchased from InvivoGen, USA. Cells were cultured in the presence of blasticidin (Nacalai Tesque, Japan) and hygromycin B (Cayman Chemical, USA) according to the manufacturer's instructions. Both HEK293 cell lines were transfected with NF-κB luciferase reporter vector and β-galactosidase gene. For the experiments using transiently murine TLR-expressing HEK293 cells, murine TLR2, TLR3, TLR7, TLR9, and Unc93B1 cDNAs were cloned into the pMX-EF1α-IRES2-Venus expression vector. Unc93B1, NF-κB luciferase reporter vector, and β-galactosidase gene were transiently cotransfected with each TLRs vector to HEK293 cells. To examine the response to DJ-1 overexpression, the DJ-1 cDNA vector that was cloned into pcDNA3.1 was cotransfected with Unc93B1, NF-κB luciferase reporter vector, β-galactosidase gene, and each of the TLRs vectors. Recombinant DJ-1 protein (1 μM) was added to the transfected HEK293 cell lines, and luciferase activities were measured using a Luciferase Assay System (Promega, USA). A total of 10 μg mL$^{-1}$ of peptidoglycan (Sigma-Aldrich, USA), 1 μg mL$^{-1}$ of

lipopolysaccharide (Sigma-Aldrich, USA), 5 µg mL$^{-1}$ of Imiquimod (TCI, Japan), 10 µM of CpG-DNA, and 10 mg mL$^{-1}$ of poly(I:C) (InvivoGen, USA) were used as positive controls for these experiments. A plasmid containing the β-galactosidase gene was used to normalize for transfection efficiency.

## Immunohistochemistry

For TUNEL staining, paraffin-embedded sections were deparaffinized and incubated with proteinase K (20 µg ml$^{-1}$) and 0.3% hydrogen peroxide. Terminal deoxynucleotidyl transferase and dUTP-biotin were added to each section and the sections were incubated at 37˚C for 1 h. Horseradish peroxidase (HRP) or iFluor 546 (AAT Bioquest, USA, 1:300 dilution, Cat# 16958) was used for the detection of dUTP-biotin. TUNEL-positive neuronal cells were counted in the peri-infarct region, which was considered to extend 3 mm lateral from the midline as previously described [33]. The counts of positive neuronal cells within 3 different areas (each 0.1 mm$^2$) were expressed as an averaged value. For the detection of extracellular DAMPs, deparaffinized sections were incubated with proteinase K (20 µg ml$^{-1}$) and were blocked with Blocking One Histo (Nacalai Tesque, Japan). Sections were incubated with primary antibody overnight at 4˚C and washed with PBS 3 times. Anti-NeuN antibody (Millipore, USA, 1:500 dilution, Cat# MAB377), anti-F4/80 antibody (Bio-Rad, USA, 1:300 dilution, Cat# MCA497RT), anti-Olig2 antibody (Millipore, USA, 1:300 dilution, Cat# AB9610), and anti-GFAP antibody (Sigma-Aldrich, USA, 1:1000 dilution, Cat# G9269) were used for the detection of neurons, macrophages, oligodendrocytes, and astrocytes, respectively. Anti-pan-cadherin antibody (Abcam, United Kingdom, 1:30 dilution, Cat# ab6529) was used for the detection of cellular membrane. Alexa488-conjugated secondary antibody (Thermo Fisher Scientific, USA, 1:300 dilution, Cat# A11070 for rabbit IgG) or Alexa546-conjugated secondary antibody (Thermo Fisher Scientific, USA, 1:300 dilution, Cat# A11081 for anti-rat IgG, Cat# A11018 for anti-mouse IgG, Cat# A11071 for anti-rabbit IgG) was used for the detection of primary antibodies. To detect protein–protein interaction, we performed PLA using a Duolink kit (Sigma Aldrich, USA) according to the manufacturer's protocol. Anti-TLR2 antibody (Abcam, USA, 1:50 dilution, Cat# ab16894) or anti-TLR4 antibody (Santa Cruz Biotechnology, USA, 1:50 dilution, Cat# sc-13593) was used for detecting PLA signals in the myeloid cells detected by anti-CD11b antibody (GeneTex, USA, 1:100 dilution, Cat# GTX26332). To quantify extracellular DJ-1-positive areas or intracellular DJ-1-positive areas (outside or inside the cellular membrane detected by pan-cadherin staining), each DJ-1-positive area in a 0.25-mm$^2$ area was measured using Fiji software (NIH, USA). To quantify areas where DJ-1-including debris contacted the cellular membranes of infiltrating myeloid cells, each DJ-1-contacted area on the cellular membranes of CD11b$^+$ cells in a 0.16-mm$^2$ area was measured using Fiji software (NIH, USA). Images of the sections were observed and captured under a fluorescence microscope (BZ-X710, Keyence, Japan) or a confocal laser microscopy (LSM710, Carl Zeiss, Germany).

## Preparation of the population enriched with infiltrating immune cells from the brain

Mice were transcardially perfused with ice-cold PBS to exclude circulating blood cells. The forebrain was homogenized in RPMI-1640 and treated with type IV collagenase (1 mg ml$^{-1}$, Sigma, USA) and DNase I (50 µg ml$^{-1}$, Sigma, USA) at 37˚C for 45 min. Infiltrating immune cells were collected from the interphase of 37/70% Percoll (GE Healthcare, USA) and used for further experiments. For fluorescence-activated cell sorting (FACS) analysis, infiltrating inflammatory cells from ischemic brain tissue were prepared using Percoll, and surface

staining was performed for 15 min with the corresponding mixture of fluorescently labeled antibodies. Infiltrating inflammatory cells were detected using the following surface markers: microglia included CD45-intermediate and CD11b-intermediate; macrophages included CD45-high, CD11b-high, F4/80[+]. FACS analysis was performed using a FACS Aria III instrument (BD Biosciences, USA) and analyzed using FlowJo software (Tree Star, USA).

## Neuronal cell death assay

Culture supernatant of activated BMMs treated with 1.0 μM of recombinant GST or DJ-1 protein for 24 h was added to the primary cultured neurons. To generate primary cultured neurons, we isolated embryonic cortical neurons from E14 embryonic cerebral cortices. Cells were suspended in a Neurobasal (Gibco, USA) medium supplemented with B27 (Gibco, USA), then plated on dishes coated with poly-L-lysine. Neuronal cell death was examined using SYTOX-green (Thermo Fisher Scientific, USA) 24 h after adding recombinant proteins or culture supernatants.

## Accession numbers

Each coding sequence used for cDNA cloning into the expression vector can be obtained from the National Center for Biotechnology Information (NCBI; gene accession number: *Park7*(DJ-1), NM_020569; *Tlr2*, NM_011905; *Tlr3*, NM_126166; *Tlr7*, NM_001290756; *Tlr9*, NM_031178; *Unc-93b1*, NM_001161428). The crystal structure of DJ-1 protein can be found in the Protein Data Bank (PDB ID: 1P5F).

## Statistical analysis

One-way analysis of variance (ANOVA) followed by post hoc multiple-comparison tests (Dunnett correction) was used to analyze differences among three or more groups of mice or samples. Between 2 groups of mice or samples, an unpaired Student $t$ test was performed to determine statistical significance. Wilcoxon rank sum test with Bonferroni correction was performed to determine statistical significance of neurological deficits. $p < 0.05$ was considered a significant difference.

## Supporting information

**S1 Fig.** The IL-23p19 mRNA expression of BMDCs treated with brain homogenate alone or costimulated with brain homogenate and 0.1–1.0 μM of recombinant DJ-1 protein. Experiments were performed in triplicate. The data underlying this figure can be found in S1 Data. BMDC, bone marrow–derived dendritic cell.
(PDF)

**S2 Fig.** The relative luciferase activity of NF-κB reporter in each transiently murine TLR-expressing HEK293 cell that was transfected with mock or DJ-1 expression vector. TLR ligands were added as a positive control. Experiments were performed in triplicate. The results are representative of 3 independent experiments; one-way ANOVA with Dunnett correction. *$p < 0.05$, **$p < 0.01$, ***$p < 0.001$ vs. mock. The data underlying this figure can be found in S1 Data. ANOVA, analysis of variance; NF-κB, nuclear factor kappa B; TLR, Toll-like receptor.
(PDF)

**S3 Fig.** IL-23p19-inducing activities of BMMs treated with DJ-1 protein or mutant DJ-1 protein (C106S) or modified DJ-1 proteins that were treated with PBS or iodo after DTT

treatment. Each mRNA expression level is shown relative to that of GST-treated BMMs. Experiments were performed in triplicate. The results are representative of 2 independent experiments; one-way ANOVA with Dunnett correction; n.s., not significant vs. DJ-1. The data underlying this figure can be found in S1 Data. ANOVA, analysis of variance; BMM, bone marrow–derived macrophage; DTT, dithiothreitol; GST, glutathione-S-transferase; iodo, iodoacetamide; PBS, phosphate-buffered saline;
(PDF)

**S4 Fig.** **(A)** Tyramide-enhanced immunohistochemical staining of DJ-1 in normal brain cells (NeuN for neurons, GFAP for astrocytes, Olig2 for oligodendrocytes, F4/80 for myeloid cells, and DAPI for nuclei). **(B)** Western blotting analysis of lysates collected from ischemic brain tissues of WT or DJ-1-deficient mice. **(C)** Immunohistochemical staining of DJ-1, NeuN, and DAPI in the infarct area of DJ-1-deficient mice 24 h after stroke onset. Scale bars: 10 μm [**A, C**]. GFAP, glial fibrillary acidic protein; WT, wild-type.
(PDF)

**S5 Fig.** **(A)** Immunohistochemical staining of DJ-1, NeuN, and DAPI in the infarct area of the permanent MCAO model. Time-dependent changes in the ischemic brain were investigated at the indicated time points. Scale bar: 10 μm. **(B)** Quantification of intracellular or extracellular DJ-1-positive areas in the infarct area 24 h after stroke onset. $n = 6$ mice for each group. Two-sided Student $t$ test; $^{***}p < 0.001$ vs. intracellular DJ-1-positive area. The data underlying this figure can be found in S1 Data. MCAO, middle cerebral artery occlusion.
(PDF)

**S6 Fig.** **(A)** Western blotting analysis of the culture supernatant (OGD sup) and cellular lysate (OGD lysate) of primary neurons subjected to the OGD. The samples were collected from 1 dish of cultured primary neurons at each time point after OGD and same liquid volumes of each sample were analyzed by western blotting. **(B)** The IL-23p19 mRNA expression of BMDCs treated with the culture supernatant of primary neurons 24 h after OGD with/without anti-DJ-1 antibody. One-way ANOVA with Dunnett correction; $^{*}p < 0.05$, $^{**}p < 0.01$ vs. BMDCs treated with the cell supernatant of primary neurons after OGD. Experiments were performed in triplicate. The results are representative of 3 independent experiments. The data underlying this figure can be found in S1 Data. ANOVA, analysis of variance; BMDC, bone marrow–derived dendritic cell; OGD, oxygen-glucose deprivation.
(PDF)

**S7 Fig.** The absolute number of CD45$^{high}$CD11b$^{high}$ population (macrophages/neutrophils) collected from the ischemic brain at each time point after the onset of ischemic stroke ($n = 4$ mice for sham, 12 h, and 3 days; $n = 3$ mice for 24 h). The data underlying this figure can be found in S1 Data.
(PDF)

**S8 Fig.** **(A)** The gating strategies for collecting macrophages/neutrophils (CD11b-high), microglia (CD11b-intermediate), and astrocytes (CD11b$^−$, CD105$^−$, CD140a$^−$, B220$^−$, O4$^−$) by FACS. This method for sorting astrocytes was described elsewhere [34,35]. **(B)** The successful isolation of astrocytes was confirmed by the mRNA expression levels of *Aldh1l1* (an astrocyte marker) in the astrocytes, microglia, and macrophages/neutrophils collected from sham-operated mice or day 1 post-ischemic brains of mice treated with DJ-1-specific antibody or control IgG antibody immediately after stroke onset. **(C)** The mRNA expression levels of TNFα in the pooled astrocytes, microglia, and macrophages/neutrophils population isolated by FACS from 3 sham-operated mice or day 1 post-ischemic brains of 3 mice treated with control IgG

antibody or DJ-1-specific antibody immediately after stroke onset. The data underlying this figure can be found in S1 Data. FACS, fluorescence-activated cell sorting; IgG, immunoglobulin G; TNFα, tumor necrosis factor alpha.
(PDF)

**S9 Fig.** **(A)** The absolute number of infiltrating macrophages (CD11b$^{high}$F4/80$^{+}$) collected from the day 3 post-ischemic brain of mice treated with vehicle or clodronate liposome ($n = 5$ mice for each group). $^{***}p < 0.001$ vs. vehicle-treated mice. The right panel shows the representative result of FACS. **(B)** The relative mRNA expression levels in the day 3 post-ischemic brain tissue of mice treated with clodronate liposome and indicated antibody ($n = 6$ mice for control IgG, $n = 7$ mice for anti-DJ-1 antibody), compared to sham-operated mice. n.s., not significant vs. control IgG. **(C)** Infarct volume on day 7 after stroke onset and neurological deficits of mice treated with clodronate liposome and indicated antibody. n.s., not significant vs. control IgG ($n = 5$ mice for control IgG, $n = 6$ mice for anti-DJ-1 antibody). Two-sided Student $t$ test [**A–C: infarct volume**]. Wilcoxon rank sum test with Bonferroni correction [**C: neurological deficits**]. The data underlying this figure can be found in S1 Data. FACS, fluorescence-activated cell sorting; IgG, immunoglobulin G.
(PDF)

**S10 Fig.** **(A)** The percentage of living primarily cultured neuronal cells that were treated for 24 h with recombinant GST or DJ-1 or culture supernatant from intact BMMs or BMMs treated with DJ-1 for 3 h. OGD was performed as a positive control of neuronal cell death (one-way ANOVA with Dunnett correction). $^{***}p < 0.001$ vs. GST-treated cells. **(B)** The percentage of living primarily cultured neuronal cells that were treated with recombinant GST or DJ-1 at each time point. Experiments were performed in triplicate [**A, B**]. The data underlying this figure can be found in S1 Data. ANOVA, analysis of variance; BMM, bone marrow–derived macrophage; GST, glutathione-S-transferase; OGD, oxygen-glucose deprivation.
(PDF)

**S1 Table.** **(A)** Kaplan–Meier survival curve for WT vs. DJ-1-deficient mice and the percent of CBF reduction after CCA and MCA occlusion that was used in **Fig 5**. **(B)** Kaplan–Meier survival curve for control IgG antibody–administered mice vs. anti-DJ-1-administered mice and the percent of CBF reduction after CCA and MCA occlusion that was used in **Fig 6**. Values are shown as the mean ± standard error of the mean [**A, B**]. The data underlying this figure can be found in S1 Data.
(PDF)

**S2 Table.** List of primers used for quantitative PCR.
(PDF)

**S1 Data.** All raw data represented in different spreadsheet tabs per figure.
(XLSX)

**S1 Raw Images.** Unedited, original blots images for Figs 3A, S4B and S6A.
(PDF)

## Acknowledgments

We thank Dr. K. Kuroda for mass spectrometry analysis of brain homogenates and Dr. R. Sato and Dr. T. Shibata for giving us technical advice for generating endosomal- TLR-expressing HEK293 cells. We also thank Dr. J. Horiuchi for the assistance with writing this manuscript and Dr. H. Shitara and Dr. R. Ishii for supporting animal breeding.

## Author Contributions

**Conceptualization:** Koutarou Nakamura, Seiichiro Sakai, Takashi Shichita.

**Data curation:** Koutarou Nakamura, Seiichiro Sakai, Jun Tsuyama, Takashi Shichita.

**Formal analysis:** Koutarou Nakamura, Seiichiro Sakai, Jun Tsuyama, Akari Nakamura, Takashi Shichita.

**Funding acquisition:** Takashi Shichita.

**Investigation:** Koutarou Nakamura, Seiichiro Sakai, Jun Tsuyama, Akari Nakamura, Kento Otani, Kumiko Kurabayashi, Yoshiko Yogiashi, Takashi Shichita.

**Methodology:** Koutarou Nakamura, Seiichiro Sakai, Jun Tsuyama, Akari Nakamura, Kento Otani, Yoshiko Yogiashi, Takashi Shichita.

**Project administration:** Takashi Shichita.

**Resources:** Takashi Shichita.

**Supervision:** Hisao Masai, Takashi Shichita.

**Writing – original draft:** Koutarou Nakamura, Takashi Shichita.

**Writing – review & editing:** Koutarou Nakamura, Seiichiro Sakai, Hisao Masai, Takashi Shichita.

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
