## [Editor Report · Decision Letter 0]

10 Sep 2020

Dear Dr Shichita, 

Thank you for submitting your manuscript entitled "Extracellular DJ-1 induces sterile inflammation in the ischemic brain" for consideration as a Research Article by PLOS Biology.

Your manuscript has now been evaluated by the PLOS Biology editorial staff as well as by an academic editor with relevant expertise and I am writing to let you know that we would be interested to consider a revision. However, before we can move forward, we need you to complete your submission by providing the metadata that is required for full assessment. To this end, please login to Editorial Manager where you will find the paper in the 'Submissions Needing Revisions' folder on your homepage. Please click 'Revise Submission' from the Action Links and complete all additional questions in the submission questionnaire.

Please re-submit your manuscript within two working days, i.e. by Sep 14 2020 11:59PM.

Once your full submission is complete, your paper will undergo a series of checks. Once your manuscript has passed all checks, we will send you a "revision decision", which will give you the opportunity to revise your manuscript as you have outlined in the current response to reviewers and then submit it to us. 

Kind regards,

Lucas Smith, Ph.D.,

Associate Editor

PLOS Biology

---

## [Editor Report · Decision Letter 1]

21 Sep 2020

Dear Dr Shichita,

Thank you very much for submitting your manuscript "Extracellular DJ-1 induces sterile inflammation in the ischemic brain" for consideration as a Research Article at PLOS Biology. As mentioned in our last email, your manuscript has been evaluated by the PLOS Biology editors and an Academic Editor with relevant expertise. 

In light of the reviews from Review Commons, we will not be able to accept the current version of the manuscript, but we would welcome re-submission of a much-revised version that takes into account the reviewers' comments as outlined in your submission. We cannot make any decision about publication until we have seen the revised manuscript and your response to the reviewers' comments. Your revised manuscript is also likely to be sent for further evaluation by the reviewers. We will do our best to engage with the same reviewers that initially reviewed the paper at Review Commons, however this is not always possible, as it depends on the reviewer availability. However, if we deviate from this commitment for external circumstances, we will let you know. 

We expect to receive your revised manuscript within 3 months. 

**IMPORTANT - SUBMITTING YOUR REVISION**

*Re-submission Checklist*

*Published Peer Review*

*PLOS Data Policy*

*Blot and Gel Data Policy*

Sincerely,

Lucas Smith, Ph.D.,

Associate Editor,

lsmith@plos.org,

PLOS Biology

---

## [Decision Letter · Decision Letter 2]

4 Feb 2021

Dear Dr Shichita,

Thank you for submitting your revised Research Article entitled "Extracellular DJ-1 induces sterile inflammation in the ischemic brain" for publication in PLOS Biology. I have now obtained advice from the original reviewers and have discussed their comments with the Academic Editor. 

As you will see, all of the reviewers think the manuscript has been substantially improved during revision and they do not have additional comments to address. Therefore, based on the reviews, we will probably accept this manuscript for publication, assuming that you will address the data and other policy-related requests noted at the end of this email.

We expect to receive your revised manuscript within two weeks. Your revisions should address the specific points made by each reviewer. 

-  a cover letter that should detail your responses to any editorial requests, if applicable

*Published Peer Review History*

*Early Version*

Sincerely,

Lucas Smith, Ph.D.,

Associate Editor,

lsmith@plos.org,

PLOS Biology

**IMPORTANT, Please address the data and policy requests listed below in a revised manuscript:

ETHICS STATEMENT:

-- Please include the specific national or international regulations/guidelines to which your animal care and use protocol adhered. Please note that institutional or accreditation organization guidelines (such as AAALAC) do not meet this requirement.

DATA POLICY:

[Figs 1A-C;2A-E; 3A,B-C; 4D-F; 5A-D; 6A-G; S1; S2; S3;S5B; S6B; S7; S8B-C; S9A-C; S10; Table S1; ]

Reviewer remarks:

Reviewer's Responses to Questions

PLOS authors have the option to publish the peer review history of their article (what does this mean?). If published, this will include your full peer review and any attached files.

Reviewer #1: Yes: Yvonne Couch

Reviewer #2: Yes: Arthur Liesz

Reviewer #3: Yes: Christoph Harms

Reviewer #1: The authors have extensively and comprehensively addressed all concerns and the manuscript is much improved. After having reviewing comments ignored by a number of authors it is refreshing to see their attitude was conversational, rather than confrontational and suggested changes were discussed in light of the literature and their own new data. Thank you for making reviewing more enjoyable than it has been recently!

Reviewer #2: The authors did an excellent job in addressing all of my concerns. I have no additional comments

Reviewer #3: The manuscript has substantially improved by the additional experiments and the transparency of the way the data are displayed. All of my concerns have vanished and I congratulate the authors to this novel and important contribution towards understanding the pathophysiology of stroke.

---

## [Editor Report · Decision Letter 3]

26 Feb 2021

Dear Dr Shichita,

Thank you for submitting your revised Research Article entitled "Extracellular DJ-1 induces sterile inflammation in the ischemic brain" to PLOS Biology. We appreciate that you have provided the supplementary files that we requested and that your manuscript now includes the underlying data presented in your figures (S1_Data) and the raw images for your western blot analysis (S1_raw_images).

We do have a few lingering data and policy-related requests which we will need you to address before we can formally accept your paper. These are fairly minor textual changes, and should not take very long to address. I have outlined them here, and included them below my signature. Please feel free to email me if you have any questions. 

1) Ethics Policy Request - Thank you for including an ethics statement in your manuscript. The ethics statement is currently missing information regarding the specific national or international regulations/guidelines to which your protocol adheres. Would you please add a sentence to your ethics statement with this information? For example, you could add the sentence saying “Experiments adhered to __ guidelines." (Fill the blank with relevant national or international guidelines). Please note that the institutional or accreditation organization guidelines (Such as AALAC) do not meet this requirement. 

2) Data Policy Request - Thank you for providing the numerical values used in your figures as S1_data. Would you please add a sentence to each figure legend that that refers to this supplemental file? For example, you could add a sentence to the end of each figure legends saying “The data underlying this figure can be found in S1_data”. Please make sure to include this sentence in the supplementary figure legends as well.

3) Please also make sure that the S1_data file contains a legend describing what it is.

We expect to receive your revised manuscript within two weeks. 

*Published Peer Review History*

*Early Version*

Sincerely,

Lucas Smith, Ph.D.,

Associate Editor,

lsmith@plos.org,

PLOS Biology

ETHICS STATEMENT:

-- Please include the specific national or international regulations/guidelines to which your animal care and use protocol adhered. Please note that institutional or accreditation organization guidelines (such as AAALAC) do not meet this requirement.

DATA POLICY:

--Please ensure that figure legends in your manuscript include information on where the underlying data can be found, and ensure your supplemental data file/s has a legend.

---

## [Editor Report · Decision Letter 4]

9 Mar 2021

Dear Dr Shichita,

On behalf of my colleagues and the Academic Editor, Richard Daneman, I am pleased to say that we can in principle offer to publish your Research Article "Extracellular DJ-1 induces sterile inflammation in the ischemic brain" in PLOS Biology, provided you address any remaining formatting and reporting issues. These will be detailed in an email that will follow this letter and that you will usually receive within 2-3 business days, during which time no action is required from you. Please note that we will not be able to formally accept your manuscript and schedule it for publication until you have made the required changes.

When addressing these last requests, we would also appreciate it if you could also update your ethics statement to include the name of the national or international animal welfare/handling guidelines to which your protocol adhered. Please note, that the ARRIVE guidelines do not seem to meet this criteria, as they are reporting guidelines. An example of national/international ethical guidelines would be the "Act on Welfare and Management of Animals of Japan" or the NIH's "guide for the care and use of laboratory animals". If you have any questions about this request, please feel free to email me.

PRESS

Thank you again for supporting Open Access publishing. We look forward to publishing your paper in PLOS Biology. 

Sincerely, 

Lucas Smith, Ph.D. 

Associate Editor 

PLOS Biology

lsmith@plos.org